# Simulations of Flows via CFD in Microchannels for Characterizing Entrance Region and Developing New Correlations for Hydrodynamic Entrance Length

**DOI:** 10.3390/mi14071418

**Published:** 2023-07-14

**Authors:** Dustin R. Ray, Debendra K. Das

**Affiliations:** Department of Mechanical Engineering, University of Alaska Fairbanks, Fairbanks, AK 99775-5905, USA; dkdas@alaska.edu

**Keywords:** microchannel, entrance length, low Reynolds number, CFD simulations

## Abstract

Devices with microchannels are relatively new, and many correlations are not yet developed to design them efficiently. In microchannels, the flow regime is primarily laminar, where entrance length may occupy a significant section of the flow channel. Therefore, several computational fluid dynamic simulations were performed in this research to characterize the developing flow regime. The new correlations of entrance length were developed from a vast number of numerical results obtained from these simulations. A three-dimensional laminar flow for 37 Reynolds numbers (0.1, 0.2, …, 1, 2, …, 10, 20, …, 100, 200, …, 1000), primarily in low regime with water flow through six rectangular microchannels (aspect ratio: 1, 0.75, 0.5, 0.25, 0.2, 0.125), has been modeled, conducting 222 simulations to characterize flow developments and ascertain progressive velocity profile shapes. Examination of the fully developed flow condition was considered using traditional criteria such as velocity and incremental pressure drop number. Additionally, a new criterion was presented based on *fRe*. Numerical results from the present simulations were validated by comparing the fully developed velocity profile, friction factor, and hydrodynamic entrance length for *Re* > 100 in rectangular channels, for which accurate data are available in the literature. There is a need for hydrodynamic entrance length correlations in a low Reynolds number regime (*Re* < 100). So, the model was run numerous times to generate a vast amount of numerical data that yielded two new correlations based on the velocity and *fRe* criteria.

## 1. Introduction

Microchannel devices, such as heat exchangers [1,2], offer compactness and high heat transfer efficiency, making them promising for the aerospace, computer, and electronics industries. Microchannels also see applications in microreactors [3,4], micromixing, bioanalytical instruments, and lab-on-a-chip. Microchannel hydraulic diameters typically range from 10 µm to 200 µm.

The demand for microchannel devices has significantly increased with the continuous advancements in military and civilian airborne and space-based electronics. To tackle the associated challenges, researchers are actively developing compact microchannel heat exchangers [1,2]. Recent research, exemplified by Bar-Cohen’s encyclopedia [5], focuses on fluid dynamics, thermal analyses, and optimization techniques for designing efficient microchannel flow passages. Additionally, researchers are exploring more efficient microchannel designs to handle high-density heat fluxes, such as two-phase flows (particle–liquid or liquid–vapor) and critical heat flux phenomena, using nanofluids in cooling systems [6,7,8]. Thome [9] provides insights into these cutting-edge research topics in the electronics sector.

Despite extensive research on two-phase fluid flow in microchannels, understanding single-phase hydrodynamic entrance length and developing friction factors remain limited, particularly in the low Reynolds number regime (*Re* < 100).

The design of microchannel sections heavily relies on accurately determining the development length, as a considerable portion of the flow typically undergoes development in microchannels. To meet this requirement, researchers must successfully predict the hydrodynamic development length. This research fulfills this need by providing accurate predictions for hydrodynamic development length.

Several papers (Table 1) provide microchannel fluid dynamic behavior insights through experimental and numerical studies. Qu et al. [10] observed reasonably good agreement between experimental and numerical velocity field results for developed and developing flows, suggesting that the Navier–Stokes equation can accurately predict liquid flow in microchannels. Sharp and Adrian [11] found that the Poiseuille relations were in good agreement for Reynolds numbers (Re) below 1800, and the transition from laminar to turbulent flow occurred around Reynolds numbers 1800–2000. Rands et al. [12] observed classical laminar flow behavior for low Reynolds numbers, transitioning between Reynolds numbers 2100 and 2500. Li et al. [13] confirmed that the *fRe* (friction factor times Reynolds number) and critical Reynolds number matched conventional macrotheory, with the critical Reynolds number falling within the range of 1700–1900. Celata et al. [14] found that Reynolds numbers above 300 agreed with the classical Hagen–Poiseuille law, while lower Reynolds numbers showed some discrepancy within experimental accuracy. Celata et al. [15] reported that Reynolds numbers below 585 agreed with the classical Hagen–Poiseuille law. Still, higher Reynolds numbers deviated from it, transitioning between Reynolds numbers of 1900 and 2500. Judy et al. [16] observed no deviation from the Stokes flow theory within experimental accuracy. Mala and Li [17] noted that smaller diameters showed departure (higher values) from conventional theory, while larger diameters agreed. They also identified an early transition from laminar to turbulent flow at Reynolds numbers between 300 and 900. Tretheway et al. [18] showed that the no-slip condition was valid for hydrophilic microchannels, but slip may occur in hydrophobic microchannels. El-Genk and Pourghasemi [19] discovered that the friction factor decreases with increasing slip length. Peng et al. [20] found the friction factor to be larger than the conventional theory, and the transition regime around 200–700 Reynolds numbers. Lastly, Ahmad and Hassan [21] found good agreement between fully developed velocity profiles and theoretical predictions, along with developed hydrodynamic entrance length correlations. From the current literature, hydrodynamic entrance length correlations for microchannels are limited, but macrochannels (conventional theory) may provide insight.

Shah and London [22] provided rectangular channels’ most comprehensive laminar flow data by compiling hundreds of references in their book *Laminar Flow Forced Convection in Ducts*. The majority of macrochannel studies were conducted in the 1960s–1970s. The most prominent research is summarized in Table 2. In Table 2, *X* denotes the parameters investigated. From Table 2, several researchers explored hydrodynamic entrance length for rectangular channels, but none agreed due to choosing various aspect ratios and fully developed criteria, as shown in Section 4.3.1. Most of the analytical investigations invoked an idealization of the boundary-layer assumption for simplicity, which is valid for *Re* ≥ 100 [23]. The boundary-layer assumption was acceptable when these studies were conducted, as low Reynolds flow was rare. However, with microchannels finding wide applications and having operating Reynolds numbers as low as 0.1, it demands that new analysis be conducted to develop new equations to determine hydrodynamic parameters in very low Reynolds number flows in microchannels. The present paper develops these parameters from a large number of simulation-based investigations.

### Research Objective

Most of the research conducted on rectangular channels occurred during 1960–1970. The researchers usually applied the boundary-layer assumption, covering the applicable laminar Reynolds number range *Re* ≥ 100 prevalent at their time. With microchannels, the applicable Reynolds number could dip as low as 0.1 for many practical cases. Since there is a lack of information, understanding microchannel fluid mechanics is promising for the future. Therefore, the present study aims to develop correlations for hydrodynamic entrance length for rectangular microchannels for aspect ratios: 1, 0.75, 0.5, 0.25, 0.2, 0.125, and 0.1 ≤ *Re* ≤ 1000. This broad Reynolds number range will cover the three types of laminar flows described by White [35]:
0≤Re≤1Highly viscous laminar “creeping” motion1≤Re≤100Laminar, strong Reynolds number dependence100≤Re≤1000Laminar, boundary-layer theory useful

The correlations were developed from a significant amount of numerical simulations using a well-known computational fluid dynamic (CFD) software, ANSYS Fluent [36] Version 18.1. The correlations are not only valid for rectangular microchannels, but also for macrochannels under a laminar flow.

## 2. Computational Fluid Dynamic Modeling

### 2.1. Geometry of the Microchannels

A three-dimensional model is shown in Figure 1a for a typical rectangular microchannel. With the assumption of uniform fluid distribution, the system can be analyzed by examining a quarter of a channel, as shown in Figure 1b, by invoking symmetry. The cross-sectional coordinates are *x* and *y*, while the axial coordinate is *z.* The origin is at the center of the channel (bottom left corner of Figure 1b).

The variable parameters for the present modeling are the Reynolds number and aspect ratio (α=a/b). Reynolds numbers will vary from 0.1 to 1000 in a logarithmic pattern, as shown in Table 3, for a total of 37 different Reynolds numbers. Six different aspect ratios are considered. The channel’s width (2*a*) is varied to create the various aspect ratios. Note that the high Reynolds numbers can produce impractically high velocities for the prescribed microchannel dimensions. Still, the authors wanted to cover a broad range of Reynolds numbers, and use traditional theory to validate the results due to limited works in the lower or more applicable ranges. Following the criteria of Mehendale et al. [37] ($1\;\mu m<D_{h}<100\;\mu m$) and Kandlikar and Grande [38] ($1\;\mu m<D_{h}<200\;\mu m$) for microchannels, the largest hydraulic diameter is 100 µm. The hydraulic diameters for each aspect ratio are shown in Table 3. The hydraulic diameter is calculated using traditional theory, Equation (1) [23]. The channel lengths varied based on the required length to determine a fully developed flow, and is further explained in more detail in Section 3.1.
(1)$$D_{h}=\frac{4A_{c}}{p}=\frac{4\cdot (2a\cdot 2b)}{2\cdot 2a+2\cdot 2b}=\frac{4\cdot a\cdot b}{a+b}$$

### 2.2. Conservation Equations

For the present study, the governing equations [23,39] are conservation of mass Equation (2) and conservation of momentum Equation (3), assuming that gravitational force is negligible, fluid is incompressible, and the flow is under steady-state condition.
(2)$$\nabla \cdot \vec{V}=\frac{\partial u}{\partial x}+\frac{\partial v}{\partial y}+\frac{\partial w}{\partial z}=0$$
(3)$$\rho \left(\vec{V}\cdot \nabla \right)\vec{V}=-\nabla P+\nabla \cdot \mu \nabla^{2}\vec{V}\;\rho \left(u\frac{\partial u}{\partial x}+v\frac{\partial u}{\partial y}+w\frac{\partial u}{\partial z}\right)=-\frac{\partial P}{\partial x}+\mu \left(\frac{\partial^{2}u}{\partial^{2}x}+\frac{\partial^{2}u}{\partial^{2}y}+\frac{\partial^{2}u}{\partial^{2}z}\right)\;\rho \left(u\frac{\partial v}{\partial x}+v\frac{\partial v}{\partial y}+w\frac{\partial v}{\partial z}\right)=-\frac{\partial P}{\partial y}+\mu \left(\frac{\partial^{2}v}{\partial^{2}x}+\frac{\partial^{2}v}{\partial^{2}y}+\frac{\partial^{2}v}{\partial^{2}z}\right)\;\rho \left(u\frac{\partial w}{\partial x}+v\frac{\partial w}{\partial y}+w\frac{\partial w}{\partial z}\right)=-\frac{\partial P}{\partial z}+\mu \left(\frac{\partial^{2}w}{\partial^{2}x}+\frac{\partial^{2}w}{\partial^{2}y}+\frac{\partial^{2}w}{\partial^{2}z}\right)$$

Additionally, the fluid flow is assumed to be laminar and the viscous heating dissipation is considered negligible, which is subsequently proven to be valid in Section 3.3.4.

### 2.3. Numerical Process

The system governing Equations (2) and (3) was solved by the control volume approach using ANSYS Fluent [36]. For the solver setting, the standard pressure-based and steady-state condition were chosen. Using the laminar model, the solution method utilized the SIMPLE scheme with spatial discretization for gradient, pressure, and momentum being least squares cell-based, second order, and second order upwind, respectively.

The models were initialized using the hybrid method, after which the residuals for continuity and velocities were closely monitored. Convergence was achieved for the simulation when the residuals were less than 10^−6^. The model solved for the three velocity components and the pressure throughout the interior computational domain. The data were then exported to MATLAB 2018a [40] for further postprocessing.

### 2.4. Applied Boundary Conditions

The model has a prescribed uniform axial velocity (w) applied at the inlet (*z* = 0 µm). The specified Reynolds number (Equation (4)) determines the velocity. The pressure outlet boundary condition was applied to the outlet. The no-slip condition (*U* = 0 m/s) is applied to the channel wall surfaces (*x* = *a*, 0 ≤ *y* ≤ *b* and *y* = *b*, 0 ≤ *x* ≤ *a*). Symmetry boundary conditions were applied at *x* = 0 µm, 0 ≤ *y* ≤ *b*, and *y* = 0 µm, 0 ≤ *x* ≤ *a*. The boundary conditions are shown in Figure 2.
(4)$$w=\frac{Re\cdot \mu }{\rho \cdot D_{h}}$$

### 2.5. Water Properties

The thermophysical properties of water at 308.15 K (35 °C) were used to simulate the fluid. The properties are summarized in Table 4.

Several simulations were performed with Al_2_O_3_-based ethylene glycol/water nanofluids of 1% and 4% volumetric concentrations. It turned out that no additional information was derived, as our modeling was a single-phase liquid model. A two-phase flow simulation of particles and liquid will show the difference, but that simulation would be much more complex and was not considered in this study.

### 2.6. MATLAB Postprocessing Analysis

The postprocessing analysis was conducted using MATLAB 2018a [40] and determined the following parameters.

The mean velocity [22] is calculated as the average of the cross-sectional plane, as shown in Equation (5), which equals the prescribed inlet axial velocity.
(5)$$U_{m}=\frac{1}{A_{c}}\int_{A_{c}}wdA_{c}$$

The mean wall shear [22] is determined by averaging the wall shear stress with respect to the perimeter of the channel (Equation (6)) at a given axial location (*z* location). In Equation (6), *x* represents the perimeter distance along the wall.
(6)$$\tau_{m,z}=\frac{1}{x}\int_{0}^{x}\tau_{x}dx$$

The local Poiseuille number [22] or Fanning friction factor times Reynolds number can be determined using Equation (7).
(7)$$Po_{z}=f_{z}Re=\frac{2\tau_{m,z}}{\rho U_{m}^{2}}Re$$

The apparent Poiseuille number [22] is calculated using the pressure drop up to the axial location, as presented in Equation (8).
(8)$$Po_{app,z}=f_{app,z}Re=\frac{\Delta P_{z}\cdot D_{h}^{2}}{2\mu \cdot U_{m}\cdot z}$$

The incremental pressure drop number [22], Equation (9), is the additional pressure loss due to momentum change and the accumulated increment in wall shear between the developing and developed flow [22].
(9)$$K(z)=\left(f_{app,z}Re-fRe\right)4Z^{*}$$
where *fRe* is the fully developed value and *Z** is the dimensionless axial position, Equation (10). In fully developed flow, the incremental pressure drop number is referred to as K(∞).
(10)$$Z^{*}=\frac{z}{D_{h}\cdot Re}$$

Dimensionless hydrodynamic entrance length is defined as
(11)$$L_{h}^{+}=\frac{L_{h}}{D_{h}\cdot Re}$$
where *L_h_* is the hydrodynamic entrance length.

The velocity ratio is the maximum velocity, *U_max_*, divided by the mean velocity, as presented by Equation (12).
(12)$$U^{*}=\frac{U_{max}}{U_{m}}$$

The maximum velocity is the centerline velocity computed at the outlet.

## 3. Numerical Computation Studies

### 3.1. Study on the Independence of Channel Length

A length independence study was performed to verify the effects of channel length on hydrodynamic entrance length for a square channel, considering a Reynolds number of 1000 as an example. The study applied the standard fully developed flow criterion of centerline velocity at 99% of *U_max_*. The velocity ratio (*U**) and the dimensionless hydrodynamic entrance length were determined for four channel lengths (10, 20, 30, and 40 mm). Comparisons were made with each preceding length to determine when the parameters are independent of the channel length, as shown in Table 5.

From the study, the velocity ratio remains mostly unaffected by the channel length, with the highest difference of 0.26%. The hydrodynamic entrance length dramatically differs between the 10 mm and 20 mm channels by 6.69%. The significant change in entrance length is due to the maximum centerline velocity being an increasing asymptotic function. The slight change to the maximum centerline velocity (0.26%) would affect the hydrodynamic entrance length. At the same time, the difference in length from 20 mm to 30 mm showed a marginal effect on hydrodynamic entrance length.

Thus, the initial study had a constant channel length of 20 mm for all aspect ratios and Reynolds numbers. During postprocessing, it was found that most elements were in the fully developed region for low Reynolds numbers and aspect ratios. Thus, results for developing regions were limited. The hydrodynamic entrance length was approximated using the previous data for each subsequent model; the channel lengths were adjusted to 3 times the hydrodynamic entrance length.

### 3.2. Study on the Mesh Independence

A mesh independence study was conducted with channel lengths determined for each model. The modeling adopted orthogonal cartesian grid lines for the *xy*-plane of the channel cross-section, with smaller elements near the wall as shown in Figure 3. A bias factor increased element size along the axial flow (*z*-axis) direction. Utilizing the bias factor, 66% of the elements were contained within the suspected developing flow regime.

A mesh study was conducted for each aspect ratio (0.125, 0.20, 0.25, 0.50, 0.75, and 1.0) with a Reynolds number of 0.1 and 1000. Using the lower and upper bounds of Reynolds numbers covers the mesh validation for the Reynolds numbers in between. Due to space limitations, only the square channel mesh (*α* = 1) is discussed here. The mesh independence study consisted of four configurations, as shown in Table 6. *X*, *Y*, and *Z* elements represent the number of elements in that direction.

Multiple parameters (Δ*P*, *L_h_*^+^, *U**, *fRe*) were computed and analyzed to determine the optimal mesh. Table 7 compares pressure drop (Δ*P*) and dimensionless entrance length (*L_h_*^+^). The difference is calculated assuming that the tightest mesh, IV, is the more accurate or the so-called “true value”.

Mesh I for a Reynolds number of 0.1 displays the highest difference for pressure drop and dimensionless entrance length of 2.08% and 1.25%, respectively. In contrast, the Reynolds number of 1000 for all four meshes agrees with less than 0.70%. These results demonstrate that other than Mesh I, other meshes are reasonably acceptable.

For evaluating the accuracies of velocity ratio (*U** = *U_max_*/*U_m_*) and fully developed *fRe,* a comparison is made with known values presented by McComas [27], 2.0962 and 14.227, respectively, as summarized in Table 8. McComas’ results were accepted as the “true value” for the difference calculation. As observed in Table 7, Mesh I showed the highest difference of 1.15% and 1.29% for *U** and *fRe*, respectively, while meshes II, III, and IV show differences of less than 0.52% and 0.55%, respectively. From Table 7 and Table 8, Mesh IV produces the most accurate results. However, the computational time required for Mesh IV is nearly 2.5 times greater than Mesh III, without significantly increasing accuracy over Mesh III. Thus, Mesh III was chosen as the mesh configuration for subsequent simulations for an aspect ratio of 1.

Additional mesh independence studies were conducted for each aspect ratio following the methodology outlined here to arrive at the optimal mesh for subsequent simulations.

### 3.3. Computational Model Validation

The computational model is validated by comparing the velocity profile, *U**, and *fRe*, with published results from analytical analysis and experimental work. The temperature rise due to viscous heating is calculated to determine if viscous heating is negligible, and validated at the end of this section to be true.

#### 3.3.1. Comparison of Velocity Profiles

The fully developed velocity profile for rectangular channels is presented by McComas [27] through Equation (13). Modifications were made to match this paper’s geometric orientation, as shown in Figure 4.
(13)$$\frac{U_{fd}}{U_{m}}=\frac{1-{\left(\frac{x}{a}\right)}^{2}+4\cdot \sum_{n=1}^{\infty }\frac{{\left(-1\right)}^{n}cosh\left(\frac{N_{m}\cdot y}{a}\right)\cdot cos\left(\frac{N_{m}\cdot x}{a}\right)}{N_{m}{}^{3}cosh\left(\frac{N_{m}}{\alpha }\right)}}{\frac{2}{3}-4\alpha \sum_{n=1}^{\infty }\frac{\tanh\left(\frac{N_{m}}{\alpha }\right)}{N_{m}{}^{5}}}$$
where $N_{m}=(2n-1)\frac{\pi }{2}$.

While Equation (13) is accepted to be accurate, it is computationally complex. Although the series converges quickly, 30 terms are required to achieve five significant digits of accuracy. Therefore, due to the complexity of Equation (13), a simpler model was proposed by Purday [42]. Natarajan and Lakshmana [43] expanded upon Purday’s model and presented Equation (14a–c), where *m* and *n* are functions of aspect ratios.
(14a)$$\frac{U_{fd}}{U_{m}}=\left(\frac{m+1}{m}\right)\left(\frac{n+1}{n}\right)\left[1-{\left(\frac{y}{b}\right)}^{m}\right]\left[1-{\left(\frac{x}{a}\right)}^{n}\right]$$
(14b)$$m=1.7+0.5\cdot \alpha^{-1.4}$$
(14c)$$\begin{array}{cc} n=2 & \mathrm{for}\;\alpha \leq \frac{1}{3}\\ n=1.9+0.3\cdot \alpha & \mathrm{for}\;\alpha \geq \frac{1}{3} \end{array}$$

The fully developed axial velocity profiles from the present numerical results, Equations (13) and (14a–c), for different aspect ratios, are presented in Figure 5, which are in excellent agreement. Note McComas and Natarajan and Lakshmana are stacked on top of each other. The velocity profiles are on the center plane where *x* = 0 (Figure 5a) and *y* = 0 (Figure 5b).

The maximum difference between the present numerical results and the analytical equations is summarized in Table 9. Equation (13) is in excellent agreement with a difference of less than 2%, while Equation (14a–c) has a significantly higher difference, especially for the *y*-axis velocity, as large as 12.82%. Upon careful data comparison, disagreement occurs near the wall (0.8 ≤ *y*/*b* ≤ 1); otherwise, 4% or less was observed in the core region.

Examining Equations (13) and (14a–c) over the entire cross-sectional plane, we observe that Equation (13) has a maximum deviation of 3.9% for the square duct, and less than 2% for other aspect ratios. While Equation (14a–c) is reasonably accurate at the core region, the extremely high difference (≤39%) occurs at the corners.

#### 3.3.2. Comparison of Velocity Ratio

The ratio of maximum velocity to mean velocity (*U**) has been presented in the literature by several researchers [25,26,27,30,32,43]. Equations (13) and (14a–c) can be reduced to determine *U** by setting *x* = *y* = 0 for the centerline velocity, as shown in Equations (15) and (16).
(15)$$\frac{U_{max}}{U_{m}}=\frac{1+4\cdot \sum_{n=1}^{\infty }\frac{{\left(-1\right)}^{n}}{N_{m}{}^{3}cosh\left(\frac{N_{m}}{\alpha }\right)}}{\frac{2}{3}-4\alpha \sum_{n=1}^{\infty }\frac{\tanh\left(\frac{N_{m}}{\alpha }\right)}{N_{m}{}^{5}}}\;\mathrm{where}\;N_{m}=(2n-1)\frac{\pi }{2}$$
(16)$$U^{*}=\frac{U_{max}}{U_{m}}=\left(\frac{m+1}{m}\right)\left(\frac{n+1}{n}\right)$$

McComas [27] provided a very comprehensive set of results for ten different aspect ratios (1 ≤ *α* ≤ 0) using Equation (15), while Sparrow et al. [32] experimentally determined the velocity ratio for aspect ratios of 0.5 and 0.2. Their experimental results agreed with the calculations presented by McComas. A comparison between the present numerical results and Equations (15) and (16) is summarized in Table 10. The current numerical computation results agree with Equations (15) and (16), with differences of less than 0.34% and 1.27%, respectively.

#### 3.3.3. Comparison of Fully Developed Friction Factor with Conventional Theory

Several studies have been devoted to the essential practical parameter, the fully developed friction factor. The most widely cited research on this topic could be by Shah and London [31], who developed a correlation for *fRe* as a function aspect ratio (0 ≤ *α* ≤ 1), shown in Equation (17).
(17)$$fRe=24\cdot (1-1.3553\cdot \alpha +1.9467\cdot \alpha^{2}-1.7012\cdot \alpha^{3}+0.9564\cdot \alpha^{4}-0.2537\cdot \alpha^{5})$$

A rigorous comparison of *fRe* has been made in Table 11 between the present numerical results and Equation (17) by Shah and London [31] for six aspect ratios. An excellent agreement is observed with the equation of Shah and London with a maximum difference of 0.32%, validating this numerical model.

#### 3.3.4. Evaluation of the Effects of Viscous Heating

Viscous heating was assumed negligible in the numerical model; therefore, validation is necessary. Conservation of energy dictates that the energy loss associated with pressure drop will be converted into thermal energy, which causes the fluid temperature to rise, assuming an adiabatic condition. The energy conversion equation is Equation (18).
(18)$$\Delta P\dot{V}=\dot{m}c_{p}\Delta T$$
where ΔP is the pressure drop, $\dot{V}$ is the volumetric flow, $\dot{m}$ is the mass flow rate, *c_p_* is the fluid’s specific heat, and ΔT is the temperature rise. Equation (18) can be solved for the temperature rise and reduced to Equation (19).
(19)$$\Delta T=\frac{\Delta P}{\rho c_{p}}$$

The maximum temperature rise for each aspect ratio would occur with the highest flow rate (*Re* = 1000). The pressure loss (ΔP) is obtained from numerical computation and knowing the thermophysical properties; the temperature rise due to viscous heating is calculated and presented in Table 12. The smallest aspect ratio (0.125) caused the most significant temperature change of 1.69 K due to the high-pressure loss, while other aspect ratios were under 1 K.

Considering a temperature change of 1.69 K, the water’s density, specific heat, and viscosity would decrease by 0.06%, 0.01%, and 3.29%, respectively. Viscosity shows some change, but is marginal. Therefore, the assumption of negligible viscous heating is valid.

## 4. Results of the Numerical Studies

### 4.1. The Criterion of Fully Developed Flow

Various researchers have presented different criteria in determining fully developed flow condition: ratio of centerline velocity to maximum velocity [24,25,26,27,29], or incremental pressure drop number to fully developed incremental pressure drop number [25,28], with ranges of these ratios between 95% and 99%. The most well-known and accepted method for determining the fully developed flow condition is when the centerline velocity achieves 99% of the maximum centerline velocity. This criterion is commonly used due to the ease of measurement from experimental work using image velocimetry [21,32]. The authors will use Equations (20)–(22) for determining the developed flow condition, which will be referred to as the velocity criterion, incremental pressure drop number criterion, and *fRe* criterion, respectively.
(20)$$\frac{U}{U_{fd}}=99\%$$
(21)$$\frac{K(\infty )}{K(z)}=99\%$$
(22)$$\frac{\tau_{m,fd}}{\tau_{m,z}}=\frac{fRe}{f_{z}Re}=99\%$$
*fRe* criterion, Equation (22), is a new criterion proposed by the authors. The criterion states that fully developed flow conditions are achieved when *f_z_Re* is 99% of *fRe*. Using the *fRe* criterion for determining fully developed flow has several merits:*fRe* is already well-established from numerous studies by researchers, while fully developed incremental pressure drop number varies between researchers.*fRe* is a vital parameter engineers need to know accurately for developing or fully developed flow to determine pressure drop or pumping power requirements. Centerline velocity is less important.*fRe* is a parameter describing the fluid–wall interaction, while centerline velocity and incremental pressure depend on the fluid–wall interaction. Thus, *fRe* will achieve fully developed flow conditions before velocity or incremental pressure drop number.

One of the challenges with using the *fRe* criterion is that experimental validation would be difficult for measuring the wall shear along the channel with accurate skin friction gages.

A comparison of three fully developed flow criteria is shown in Figure 6 for a square cross-section channel with a Reynolds number of 1000. The velocity and *fRe* criteria are in reasonable agreement, while the incremental pressure drop number criteria differ greatly. The dimensionless entrance length for velocity and *fRe* criteria are 0.0733 and 0.0698 at 99%, respectively. Between the two criteria, a difference of 4.72% is observed.

The velocity and *fRe* criteria were used in all modeling cases to determine the hydrodynamic entrance length. The incremental pressure drop number did not agree with the other criteria, so it was not investigated further. Comparing the results, the difference between them increased with decreasing Reynolds number and aspect ratio.

#### 4.1.1. Reynolds Number Influence on Entrance Length

After observing how the criteria compare for a given model of an aspect ratio of 1, a comparison of the velocity and *fRe* criteria for the entire Reynolds number range is shown in Figure 7. It is a logarithmic plot with a relative difference plot included. The difference between the two criteria increases as the Reynolds number decreases until a Reynolds number of 7, with a maximum difference of 8.06%, which begins to decrease to 5.92% at a Reynolds number of 0.1.

Figure 8 compares similarly to Figure 7, but with the smallest aspect ratio of 0.125. In this case, the most significant difference is at Reynolds number 3, producing a difference of 64%.

#### 4.1.2. Aspect Ratio Effect on Entrance Length

The aspect ratio also affects the two criteria, as illustrated in Figure 9. At an aspect ratio of 1, the difference is a mere 4%. Still, as the aspect ratio decreases, the difference between the two criteria increases until an aspect ratio of 0.2, then a slight decrease occurs at 0.125.

A summary of the maximum difference for the different aspect ratios, and the Reynolds number at which it occurs, is presented in Table 13.

The entrance length calculated by the velocity criterion always comes out longer than the *fRe* criterion. The main conclusion of this investigation is that low Reynolds number and aspect ratio channels need additional studies to establish the cause of the difference. One possible reason for the differences between the velocity and *fRe* criteria may be the velocity overshoot, which is discussed below.

### 4.2. Velocity Overshoot Observation

Velocity overshoots where the centerline velocity is a local minimum, while local maxima occur near the walls, creating a concave velocity profile instead of a convex one. Velocity overshoots have been studied in detail for circular and parallel plate channels, but to the authors’ knowledge, very little has been done for rectangular channels.

Figure 10 shows an example of velocity overshoot created from the present numerical results using an aspect ratio of 1, with a Reynolds number of 1000 (Figure 10a) and Reynolds of 0.1 (Figure 10b). The “X” mark notes the maximum velocity point for that axial location.

In Figure 10a,b, a uniform velocity profile boundary condition is applied at the inlet. Moving down axially, the fluid on the wall obeys the no-slip condition with zero velocity. Thus, the fluid near it accelerates to preserve continuity, creating the overshoot. Moving axially, the velocity overshoot diminishes while converging toward the centerline. From the studies of previous researchers, velocity overshoots are negligible at high Reynolds numbers, and significant at lower Reynolds numbers. Our computational results also found similar behavior, and decreasing the aspect ratio made velocity overshoot more significant.

In Figure 10a, the velocity overshoot starts as the fluid interacts with the wall at the inlet (Z^+^ = 0). Near this location, the velocity overshoot is 1.73% greater than the centerline velocity. The velocity in the overshoot increases until Z^+^ = 2.136 × 10^−4^, showing a velocity of 10.72% greater than the centerline velocity. However, the maximum overshoot for this cross-section occurs near the corner of the duct where two wall surfaces interact with the fluid, creating a velocity 20.21% greater than the centerline velocity, as shown with a contour plot in Figure 11. From this point (Z^+^ ≥ 2.136 × 10^−4^), the overshoot diminishes as it converges towards the centerline. This happens rapidly for high Reynolds numbers, whereas in Figure 10a, the overshoot converges on the centerline at Z^+^ = 1.145 × 10^−2^. This is roughly 16.40% of the developing region. The effects of a low Reynolds number (*Re* = 0.10) on velocity overshoot are displayed in Figure 10b. Here, we notice that the velocity difference is lower than that for *Re* = 1000 of 5.57% at Z^+^ = 0.285 and 15.29% in the corner. The overshoot converges towards the centerline at Z^+^ = 1.602, about 24.08% of the developing region.

The aspect ratio has a significant effect on the velocity overshoot. This has been examined carefully from our numerical results, and displayed in Figure 12 and Figure 13 as a function of the Reynolds number.

Figure 12 shows that as the aspect ratio decreases, the difference between the overshoot and centerline velocity decreases. From Figure 12, three zones can be roughly observed: (i) from *Re* = 10^−1^ to 10, the velocity difference is the smallest and is essentially constant; (ii) from *Re* = 10 to 10^2^, the velocity difference increases and reaches a peak; and (iii) from *Re* = 10^2^ to 10^3^, the velocity difference decreases and attains a similar magnitude as the low Reynolds number flows (10^−1^ to 10).

Figure 13 shows that the aspect ratio and Reynolds number are inversely related to the distances (Z^+^) for velocity overshoot to converge towards the centerline. Thus, for a low aspect ratio, the velocity overshoot distance is higher, while the overshoot is lower. On close analysis, it was found that for an aspect ratio of 0.125, while the velocity overshoot starts in the corner, it quickly moves to the *y*-axis of symmetry. In this case, the velocity overshoot takes a significant distance to converge toward the centerline. This delay of centerline velocity achieving maximum velocity is why the velocity criterion predicts a longer entrance length than the *fRe* criterion. Thus, the authors believe that the *fRe* criterion is more suitable for determining fully developed flow for engineering applications that can accurately predict pressure drop and convective heat transfer coefficient.

### 4.3. Determination of the Hydrodynamic Entrance Length

#### 4.3.1. Comparison with Previous Correlations

Several prominent authors have researched entrance length for rectangular channels, but most results differ, and there is no universal agreement, as shown in Table 14. All of the analytical studies [24,25,26,27] applied the boundary layer theory, valid for Reynolds number 100 or greater. Beavers et al. [28] performed an experimental study on sixteen aspect ratios (1/51 to 1) with Reynolds numbers varying from 400 to 3000. They determined the dimensionless entrance length for a square channel was 0.03, and for channels with an aspect ratio less than 0.5, the dimensionless entrance length was 0.015. Goldstein and Kried [29] measured a square duct’s developing flow velocity profile using a laser-doppler flowmeter. Most of this work was performed in the 1960s and 1970s for laminar flow in channels, which would be greater than 100 for most “macro” channels. However, the present-day microchannels operate in the 0.1 ≤ *Re* ≤ 1000 range.

Recently, Ahmad and Hassan [21] conducted microparticle image velocimetry experiments for three square microchannels (500 µm, 200 µm, and 100 µm) over a Reynolds number range of 0.5 to 200. They developed an entrance length correlation, Equation (23), as a function of the Reynolds number. The correlation was within 15% of their data for square channels with a hydraulic diameter of less than 500 µm.
(23)$$\frac{L_{h}}{D_{h}}=\frac{0.63}{0.035Re+1}+0.0752Re$$

A comparison of hydrodynamic entrance length is made in Figure 14 between the numerical results of the present computation based on velocity and *fRe* criteria of flow development, and the correlations of Ahmad and Hassan [21], Han [26], and Wiginton and Dalton [24] for a square channel (*α* = 1). For better visualization, Figure 14 contains three plots expanding across a Reynolds number: (a) 0.1 ≤ *Re* ≤ 1, (b) 1 ≤ *Re* ≤ 100, and (c) 100 ≤ *Re* ≤ 1000.

Our numerical results based on the *fRe* criterion are in good agreement with Ahmad and Hassan, with a difference of less than 8% (Figure 14a,c), except for Reynolds numbers of 3–30, which increases to 10–21% observed in Figure 14b. The numerical results agreed with Han’s results for Reynolds numbers greater than 30, with a difference of less than 7.5%. Note that Ahmad and Hassan and Han overlap for large Reynolds numbers over 100. This is due to Ahmad and Hassan using Han’s results in developing their correlation for large Reynolds numbers. The results of Wiginton and Dalton differed more, with the present numerical prediction showing a minimum difference of 9.78% at a Reynolds of 20.

Similarly, our numerical results of entrance length based on velocity criteria for flow development agree with Ahmad and Hassan’s correlation, with a maximum difference of 12%. The numerical results agree with Han’s results for Reynolds numbers greater than 60, with a difference of less than 3%. The results of Wiginton and Dalton comparatively disagreed more, with numerical results showing a minimum difference of 3.4% at Reynolds number 20, while the rest were above 10%.

Figure 15 compares the dimensionless entrance length with varying aspect ratios between the present numerical results (*Re* = 1000) and Han [26], Wiginton and Dalton [24], McComas [27], and Fleming and Sparrow [25]. As evident from Figure 15, many previous researchers significantly disagree on the entrance length for rectangular channels. From the present research, it can be emphasized that the *fRe* criterion matches closely (maximum difference: 10%, mean difference: 4.17%) with Han’s entrance length prediction. The *fRe* criterion also seems to show a smooth trend with varying aspect ratios, while the velocity criterion shows a drastic trend change for aspect ratios less than 0.2.

#### 4.3.2. An Improved Correlation Was Derived

From the wealth of numerical data generated by numerous simulations for Reynolds numbers varying from 0.1 to 1000 and aspect ratios ranging from 0.125 to 1.0, two correlations for entrance length were developed: one for the velocity criterion, and another for the *fRe* criterion. A form presented by Chen [44] for circular and parallel plates, Equation (24), which Ahmad and Hassan [21] also recently adopted for rectangular channels, was selected.
(24)$$\frac{L_{h}}{D_{h}}=\frac{A}{B\cdot Re+1}+C\cdot Re$$

In Equation (24), only coefficient *B* was determined from the curve fit. The lower and upper bounds of the data determined coefficients *A* and *C*. Coefficient *A* is *L_h_*/*D_h_* at the Reynolds number of 0.1, and coefficient *C* is *L_h_^+^* at the Reynolds number of 1000.

After substantial data analysis using several MATLAB [40] scripts, the correlation coefficients *A*, *B*, and *C* were determined, listed in Table 15. Comparisons of the present numerical data with the proposed correlation are shown in Figure 16. The correlation has a good fit, with a maximum error of 10.28% for an aspect ratio of 0.2, and a mean error of less than 5%. The error means the difference between numerical data and Equation (24). The coefficients *A*, *B*, and *C* yield excellent *R*^2^ values. This table is based on data from all Reynolds numbers studied. The highest error for each aspect ratio usually occurred in the Reynolds number range of 3 to 200, which can be observed as a transition regime from a low Reynolds number (≤1) to a high Reynolds number (≥100) in Figure 16.

In Table 16, the coefficients *A*, *B*, and *C* were determined for the *fRe* criterion, which exhibited an excellent agreement, with a maximum error of 4.39% for an aspect ratio of 0.5. Figure 17 compares the numerical and correlation results using the coefficients of Table 16.

Using the coefficients *A*, *B*, and *C* determined in Table 16, correlations were developed as a function of aspect ratio. This allows the individual to determine the entrance length for any aspect ratio without interpolating between the data points. For fluid mechanics, the aspect ratio orientation (long side versus short side) for rectangular channels is irrelevant; a channel with an aspect ratio of 0.5 is the same as 2. Thus, the authors wanted to include the “symmetry” nature of the aspect ratio. The “symmetry” nature of the aspect ratio is easily noticeable once the aspect ratio’s natural log is considered, as shown in Figure 18. Figure 18a shows the dimensionless entrance length for the Reynolds number of 1000 across a broad range of aspect ratios, while Figure 18b is a similarly drawn plot with the *x*-axis as ln(*α*). In Figure 18b, the axis of symmetry can be observed at an aspect ratio of 1 (ln(1) = 0). Additionally, using the aspect ratio’s natural log, the aspect ratio’s lower and upper bounds define as parallel plates, 0 and ∞ approach −∞ and ∞, respectively. Thus, mathematically, it is easier to include parallel plates. Chen’s [44] coefficients for Equation (24) presented by Shah [22] are *A* = 0.315, *B* = 0.0175, and *C* = 0.011.

The correlations developed by including ln(*α*) are presented as Equation (25) to Equation (27). Hyperbolic secant functions can represent such symmetric curves. However, Equations (25) and (26) can be reduced to rational polynomials for ease of calculation. For Equation (27), an easy simplification was not possible due to the natural log of the aspect ratio being squared.
(25)$$A=0.437\mathrm{sech}\left(-1.07\ln(\alpha )+1.55\right)+0.437\mathrm{sech}\left(1.07\ln(\alpha )+1.55\right)+0.3125\;=\frac{4.30\alpha^{-1.07}+4.30\alpha^{1.07}}{\alpha^{-2.14}+\alpha^{2.14}+22.2}+0.3125$$
(26)$$B=0.0803\mathrm{sech}\left(2.04\ln(\alpha )\right)+0.0175\;=\frac{0.1606}{\alpha^{2.04}+\alpha^{-2.04}}+0.0175$$
(27)$$C=0.0117\mathrm{sech}\left({\left(\ln(\alpha )\right)}^{2}-0.37\right)+0.0481\mathrm{sech}\left(\ln(\alpha )\right)+0.011$$

Using Equation (24) to Equation (27), one can determine the entrance length for channels with an aspect ratio 0 ≤ *α* ≤ ∞. These correlations have a maximum error of 5.75% and a mean error of 1.87% from the numerically computed data. The relative errors are calculated for several Reynolds number and aspect ratios, and are plotted in Figure 19.

## 5. Conclusions

The compilation of earlier published research shows that most studies were on the macrochannel with boundary layer assumption, which may hold for *Re* ≥ 100. However, most microchannels will operate under *Re* ≤ 100, and possibly as low as the Reynolds number of 0.1. Therefore, new correlations for determining the hydrodynamic entrance length were necessary. Numerous numerical runs covered Reynolds numbers from 0.1 to 1000, and aspect ratios varying from 0.125 to 1. From the extensive numerical results, an investigation into the fully developed criterion was explored, examining three criteria: velocity, incremental pressure drop number, and *fRe*. The velocity and *fRe* criteria were in reasonable agreement for an aspect ratio greater than 0.5 and a Reynolds number greater than 100. The difference between the two criteria for low aspect ratio and Reynolds number may be due to the velocity overshoot. The incremental pressure drop number criterion disagreed with the other two. Thus, this criterion was not pursued any further. Hydrodynamic entrance length correlations were developed for each aspect ratio’s velocity and *fRe* criteria. The velocity criterion correlation (Equation (24) and Table 15) has a mean relative error of less than 5%, and a maximum relative error of 10.28%. The *fRe* criterion yielded a much better correlation (Equation (24) and Table 16), with a mean relative error of less than 3% and a maximum relative error of 4.39%. Additionally, the *fRe* criterion correlation was expanded to be a function of aspect ratio using Equation (24) through Equation (27), which is valid for 0.1 ≤ *Re* ≤ 1000 and 0 ≤ *α* ≤ ∞, providing a mean relative error less than 2% and a maximum relative error of 5.75%.

Future research must be focused on developing new correlations for friction factors in the entrance region of microchannels.

## Figures and Tables

**Figure 1 micromachines-14-01418-f001:**
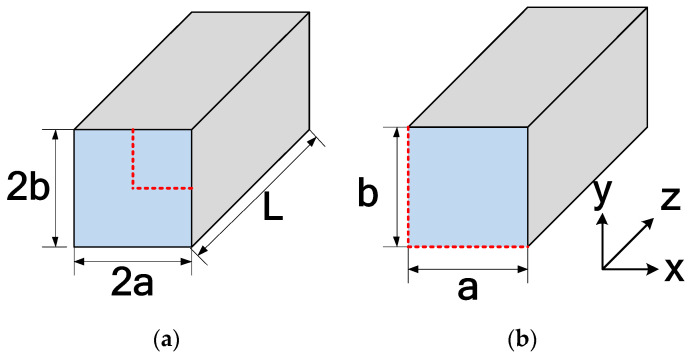
(**a**) Typical microchannel configuration used in heat sinks; (**b**) computational model.

**Figure 2 micromachines-14-01418-f002:**
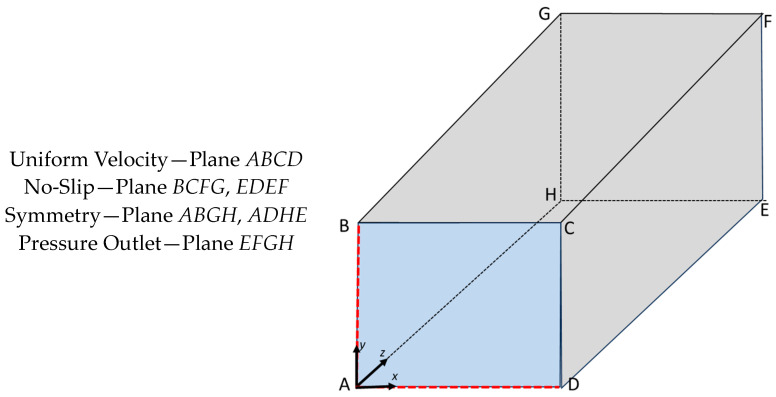
Boundary conditions.

**Figure 3 micromachines-14-01418-f003:**
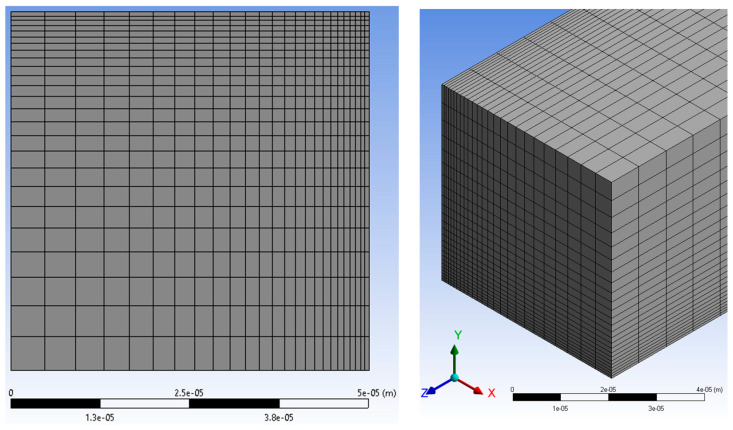
Mesh layout.

**Figure 4 micromachines-14-01418-f004:**
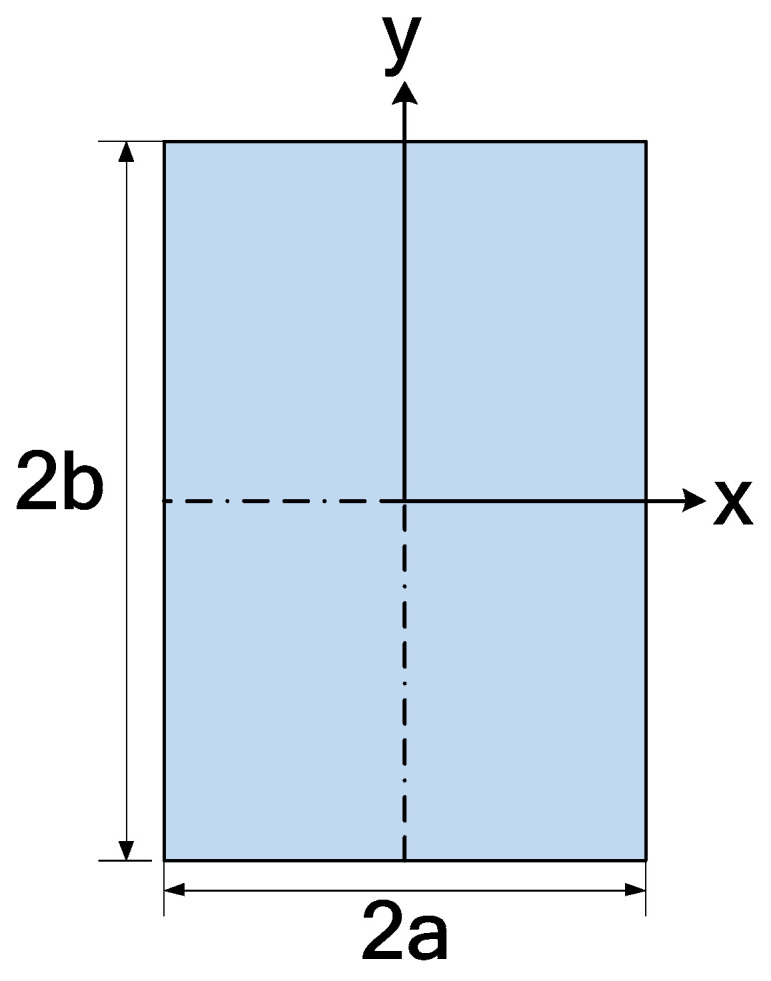
Specification of the dimensions of a rectangular duct.

**Figure 5 micromachines-14-01418-f005:**
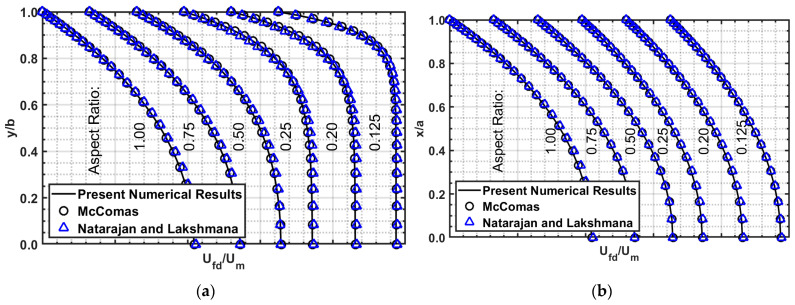
Axial velocity profile comparison for various aspect ratios at outlet between present numerical computations and theories of McComas (Equation (13)) and Natarajan and Lakshmana (Equation (14a–c)). (**a**) center plane where *x* = 0; (**b**) center plane where *y* = 0.

**Figure 6 micromachines-14-01418-f006:**
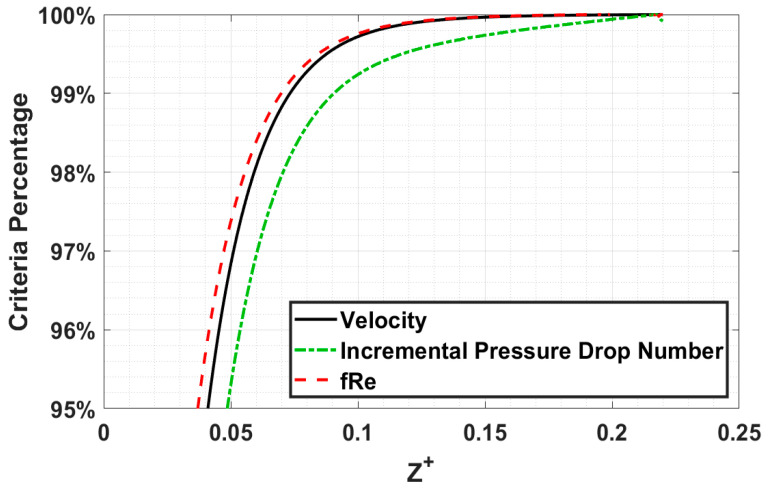
Comparison of fully developed criteria for *α* = 1 and *Re* = 1000.

**Figure 7 micromachines-14-01418-f007:**
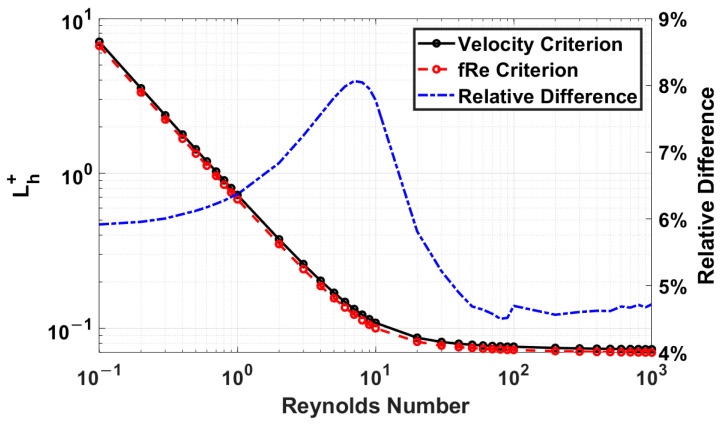
Dimensionless entrance length comparison for velocity and *fRe* criteria for an aspect ratio of 1.

**Figure 8 micromachines-14-01418-f008:**
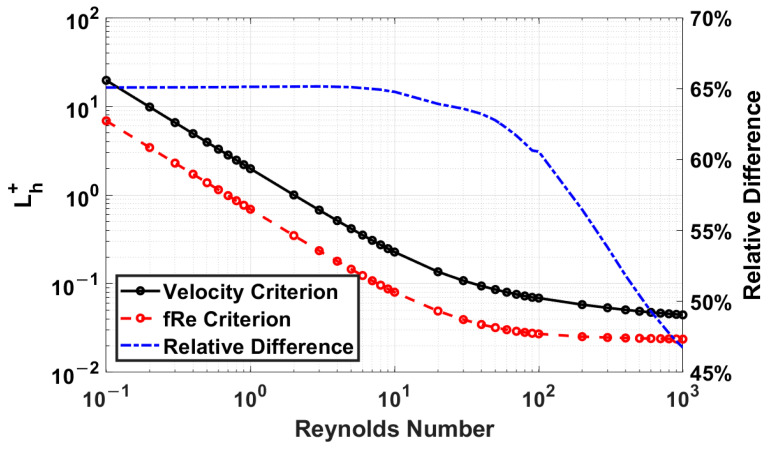
Dimensionless entrance length comparison of velocity and *fRe* criteria for an aspect ratio of 0.125.

**Figure 9 micromachines-14-01418-f009:**
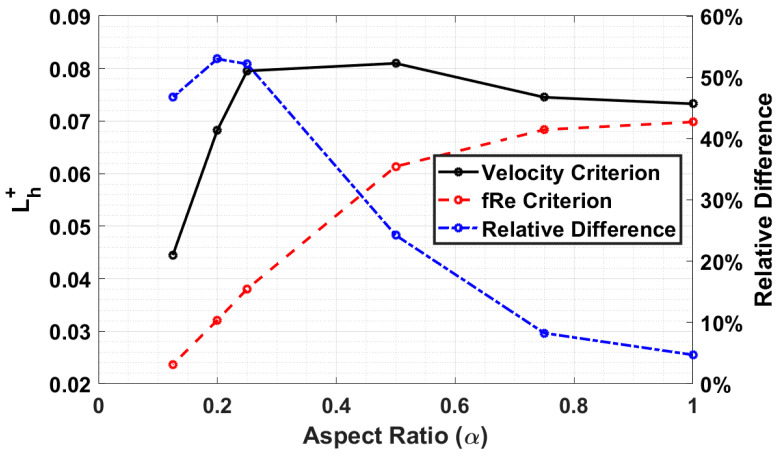
Comparing the effects of aspect ratio on dimensionless entrance length for a Reynolds number of 1000.

**Figure 10 micromachines-14-01418-f010:**
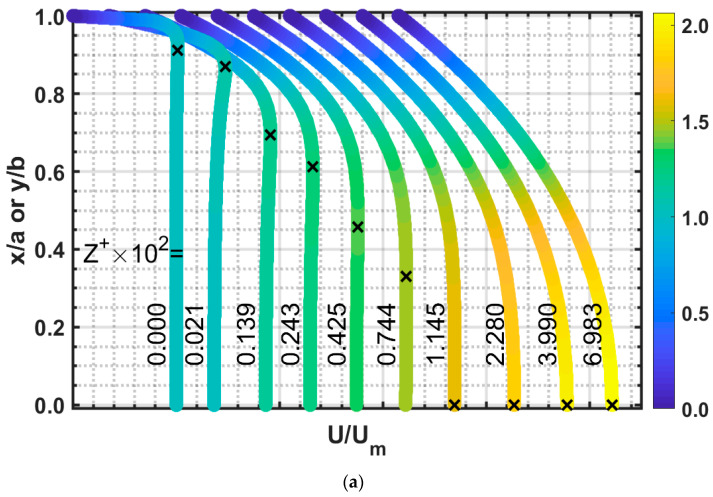
Developing velocity profile for *α* = 1, “X” mark notes the maximum velocity point for that axial location. (**a**) *Re* = 1000, (**b**) *Re* = 0.1.

**Figure 11 micromachines-14-01418-f011:**
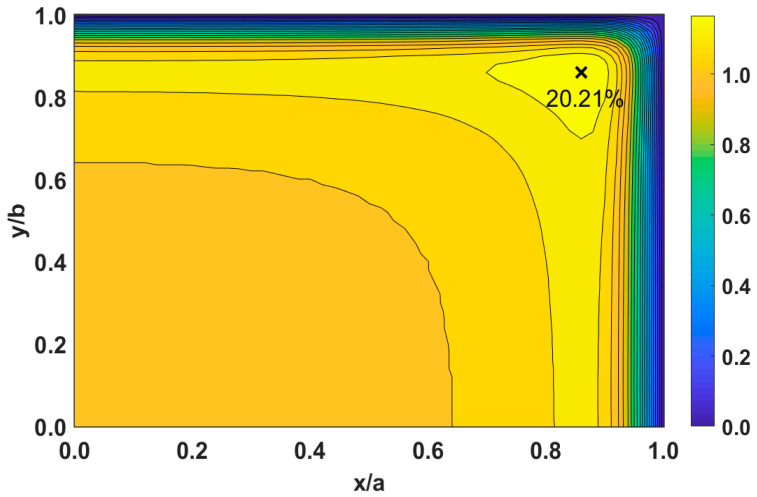
Contour plot with velocity overshoot in the corner at Z^+^ = 2.136 × 10^−4^ for an aspect ratio of 1 with Reynolds of 1000. “X” marks the location where the maximum velocity is greater than the centerline velocity by 20.21%.

**Figure 12 micromachines-14-01418-f012:**
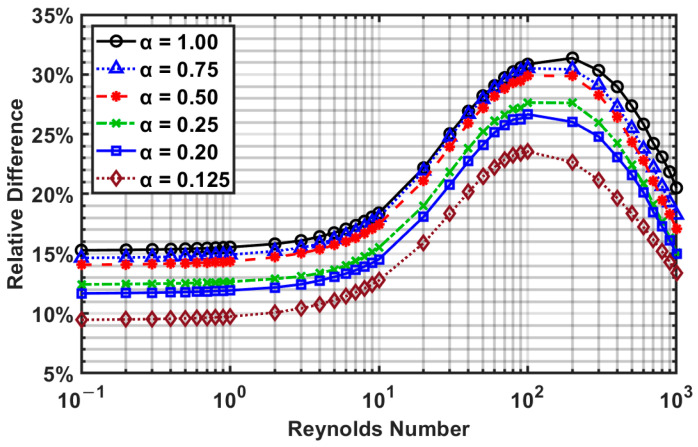
The maximum relative difference between the overshoot and centerline velocity for various aspect ratios as a function of the Reynolds number.

**Figure 13 micromachines-14-01418-f013:**
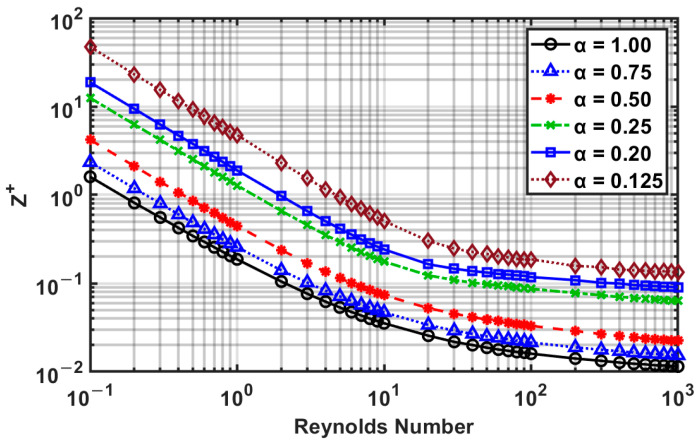
Distance for velocity overshoot to end for various aspect ratios as a function of Reynolds number.

**Figure 14 micromachines-14-01418-f014:**
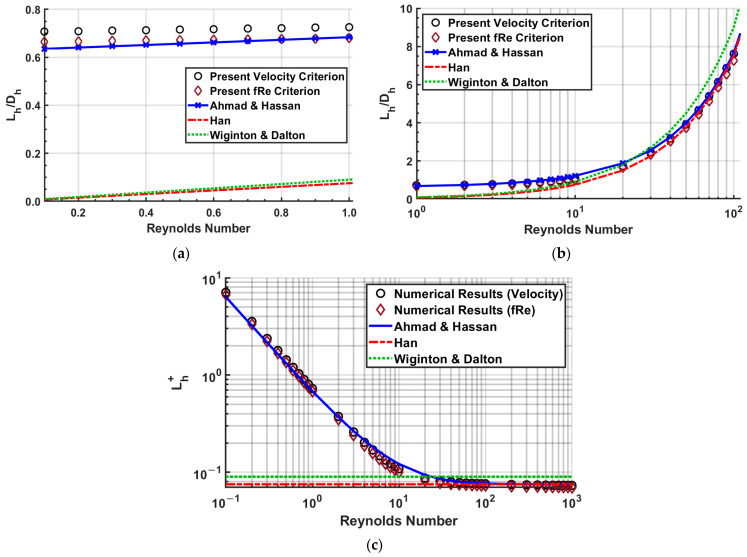
Entrance length comparisons for *α* = 1 (**a**) 0.1 ≤ *Re* ≤ 1, (**b**) 1 ≤ *Re* ≤ 100, and (**c**) 100 ≤ *Re* ≤ 1000.

**Figure 15 micromachines-14-01418-f015:**
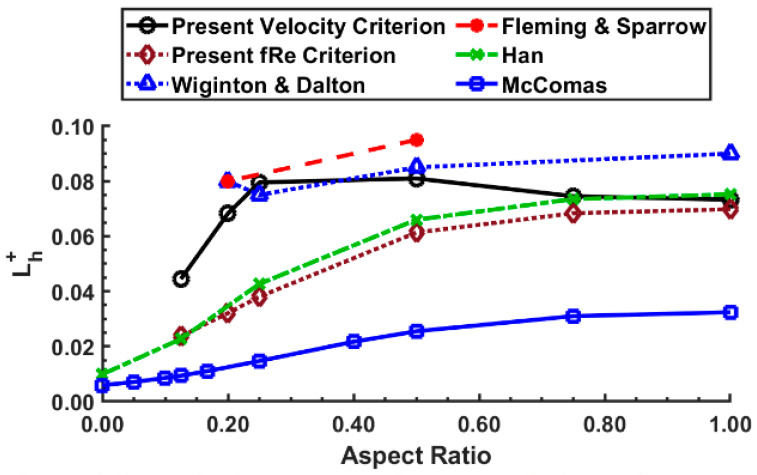
Comparison of dimensionless entrance length predictions of several authors for various aspect ratios for *Re* = 1000.

**Figure 16 micromachines-14-01418-f016:**
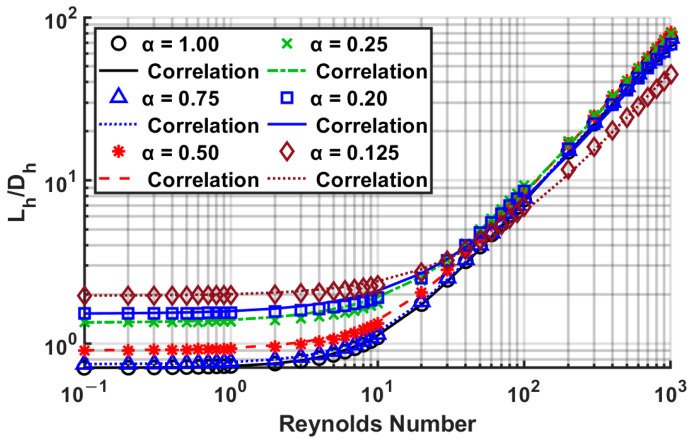
Comparison of correlation Equation (24) using Table 15 coefficients with numerically computed results.

**Figure 17 micromachines-14-01418-f017:**
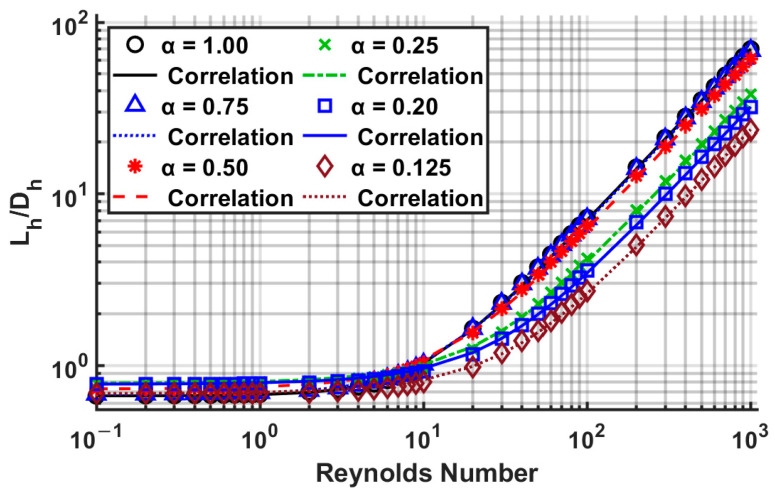
Comparison of correlation Equation (24) using Table 16 coefficients with numerically computed results.

**Figure 18 micromachines-14-01418-f018:**
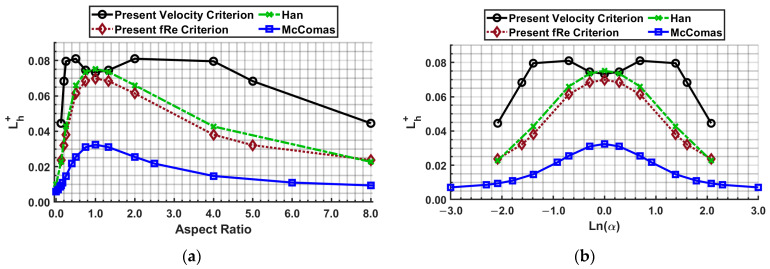
Introduction of natural logarithmic of aspect ratio to demonstrate symmetry of results (**a**,**b**).

**Figure 19 micromachines-14-01418-f019:**
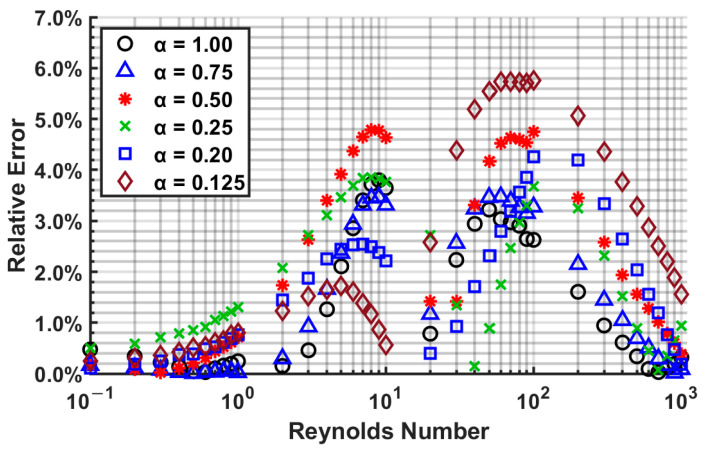
The relative error in hydraulic entrance length prediction via Equation (24) to Equation (27) from numerically computed data.

**Table 1 micromachines-14-01418-t001:** Summary of characteristics of channel, fluids, Reynolds number range, and conclusions from recent microchannel studies.

Author	Analysis	Channel Material	Channel Geometry	D_h_ (µm)	ε/D_h_	Fluid	*Re*	Observation
Qu et al. [10]	Experimental, numerical	Acrylic	Rectangular (*α* ≈ 0.32)	≈336.4	NA	Deionized water	196–2215	Reasonably good agreement between experimental and numerical results for velocity fields in developed and developing flow. Navier–Stokes equation can accurately predict liquid flow in microchannels.
Sharp and Adrian [11]	Experimental	Fused silica	Microtubes	50–247	NA	Deionized water, 1-propanol, 20/80 EG/W	20–2900	Poiseuille relations were in good agreement for *Re* < 1800. The onset of transition occurs at Reynolds number ~1800–2000.
Rands et al. [12]	Experimental	Fused silica	Microtubes	16.6–32.2	0.03–0.004% *	Deionized water	300–3400	Classical laminar flow behavior was observed for low Reynolds numbers. Results show the transition occurred at Reynolds number 2100–2500.
Li et al. [13]	Experimental	Glass	Microtubes	79.9–166.3	Smooth	Deionized water	~600–2500	*fRe* and Critical Reynolds number matched conventional macrotheory.
		Silicon	Microtubes	100.25–205.3	Smooth	Deionized water	~700–2500	
		Stainless steel	Microtubes	128.76–179.8	3–4%	Deionized water	~500–2500	D = 179.8: *fRe* increased by 15%. D = 136.5 and 128.76: *fRe* increased by 35%. Critical Reynolds in the range of 1700–1900.
Celata et al. [14]	Experimental	Smooth glass, fused silica	cCpillary tubes	31–259	NA	Degassed water	40–3000	*Re* > 300 agrees with classical Hagen–Poiseuille law. The lower *Re* number showed some discrepancy, but was within the experimental accuracy (19%—31 µm).
Celata et al. [15]	Experimental	Stainless steel	Capillary	130	2.65%	Refrigerant R114	100–8000	*Re* < 585 agrees with classical Hagen–Poiseuille law. The higher *Re* number departed from Hagen-Poiseuille law. The transition occurs from *Re* numbers 1900–2500.
Judy et al. [16]	Experimental	Fused silica, stainless steel	Round, square	15–150	NA	Distilled water, methanol, isopropanol	8–2300	No deviation from Stokes flow theory was observed within the experimental accuracy.
Mala and Li [17]	Experimental	Stainless steel, fused silica	Microtubes	50–254	1.75 µm	Water	~20–2100	Smaller diameters showed departure (higher values) from conventional theory, while larger diameters were in good agreement. Early transition 300–900.
Ahmad and Hassan [21]	Experimental	Borosilicate glass	Square	100, 200, 500	NA	Distilled water	0.5–200	Fully developed velocity profiles were in good agreement with the theory. Developed hydrodynamic entrance length correlations.

* May be considered smooth.

**Table 2 micromachines-14-01418-t002:** Summary of previous macrochannel studies.

Author	Analysis	*L_h_* ^+^	*fRe*	$\frac{U_{max}}{U_{m}}$	*K*(∞)	*f_app_Re*
Wiginton and Dalton [24]	Analytical	*X*			*X*	
Fleming and Sparrow [25]	Analytical	*X*		*X*	*X*	
Han [26]	Analytical	*X*	*X*	*X*	*X*	
McComas [27]	Analytical	*X*		*X*	*X*	
Beavers et al. [28]	Experimental	*X*			*X*	
Goldstein and Kried [29]	Experimental	*X*				
Lundgren et al. [30]	Analytical		*X*		*X*	
Shah and London [31]	Analytical		*X*			
Sparrow et al. [32]	Experimental			*X*		
Miller and Han [33]	Analytical				*X*	
Curr et al. [34]	Numerical					*X*

**Table 3 micromachines-14-01418-t003:** Summary of parameters used in the numerical simulations for microchannels.

Parameters	Symbols	Values					
Reynolds Number	Re	0.1, 0.2, …, 1, 2, …, 10, 20, …, 100, 200, …, 1000					
Hydraulic Diameter	*D_h_*	100	85.71	66.67	40.00	33.33	22.22
Aspect Ratio	α=a/b	1	0.75	0.5	0.25	0.2	0.125
Model Width (μm)	a	50	37.5	25	12.5	10	6.25
Model Height (μm)	b	50					
Length of Channel (mm)	L	Varied					

**Table 4 micromachines-14-01418-t004:** Properties of water at 308.15 K.

Property	Water [41]
Density (kg/m^3^)	994
Viscosity (mPa·s)	0.719

**Table 5 micromachines-14-01418-t005:** Channel length parameter comparisons for *Re* = 1000 and *α* = 1.

Comparing	*U** Difference	*L_h_*^+^ Difference
10 mm and 20 mm	0.26%	6.69%
20 mm and 30 mm	0.07%	0.14%
30 mm and 40 mm	0.00%	0.14%

**Table 6 micromachines-14-01418-t006:** Different mesh configurations for a square channel (*α* = 1).

Mesh	*X* Elements	*Y* Elements	*Z* Elements	Total Elements
I	15	15	200	45,000
II	20	20	300	80,000
III	25	25	400	250,000
IV	30	30	500	450,000

**Table 7 micromachines-14-01418-t007:** Mesh independence study results (Δ*P* and *L_h_*^+^) for an aspect ratio of 1.

Mesh	*Re*	Δ*P* (Pa)	Mesh IV Δ*P* (Pa)	Difference	*L_h_* ^+^	Mesh IV *L_h_*^+^	Difference
I	0.1	0.4835	0.4938	2.08%	7.0471	7.1360	1.25%
II	0.1	0.4870		1.37%	7.0495		1.21%
III	0.1	0.4900		0.77%	7.0783		0.81%
I	1000	361,620	364184	0.70%	0.0735	0.0731	0.60%
II	1000	363,099		0.30%	0.0732		0.12%
III	1000	363,785		0.11%	0.0731		0.03%

**Table 8 micromachines-14-01418-t008:** Mesh independence study results for an aspect ratio of 1 compared with the theoretical values of McComas [27].

Mesh	*Re*	*U** Difference	*fRe* Difference
I	0.1	1.15%	1.20%
II	0.1	0.51%	0.55%
III	0.1	0.28%	0.30%
IV	0.1	0.17%	0.19%
I	1000	1.17%	0.97%
II	1000	0.52%	0.43%
III	1000	0.29%	0.23%
IV	1000	0.19%	0.13%

**Table 9 micromachines-14-01418-t009:** The maximum axial velocity difference between the present scheme’s numerical results and Equations (13) and (14a–c).

Aspect. Ratio	Equation (13) *x* = 0	Equation (14a–c) *x* = 0	Equation (13) *y* = 0	Equation (14a–c) *y* = 0
1.00	1.52%	4.69%	1.79%	4.69%
0.75	1.40%	5.64%	1.52%	2.51%
0.50	1.45%	8.48%	1.52%	1.67%
0.25	1.41%	12.82%	1.50%	1.16%
0.20	1.38%	12.62%	1.48%	1.30%
0.125	1.28%	8.36%	1.47%	0.90%

**Table 10 micromachines-14-01418-t010:** Comparison of *U** results obtained from numerical computation with Equations (15) and (16).

Aspect Ratio	Numerical Results	Equation (15)	Difference	Equation (16)	Difference
1.00	2.0892	2.0963	0.34%	2.1157	1.25%
0.75	2.0713	2.0774	0.29%	2.0713	0.00%
0.50	1.9862	1.9918	0.28%	1.9805	0.29%
0.25	1.7689	1.7737	0.27%	1.7895	1.15%
0.20	1.7102	1.7150	0.28%	1.7322	1.27%
0.125	1.6235	1.6283	0.29%	1.6377	0.87%

**Table 11 micromachines-14-01418-t011:** Fully developed *fRe* comparison with Equation (17) by Shah and London [31].

Aspect Ratio	Present Numerical Results	Shah and London [31]	Difference
1.00	14.188	14.230	0.31%
0.75	14.441	14.478	0.27%
0.50	15.511	15.557	0.32%
0.25	18.187	18.234	0.27%
0.20	19.021	19.072	0.28%
0.125	20.526	20.590	0.32%

**Table 12 micromachines-14-01418-t012:** Temperature change calculated using Equation (19) for water at 308.15 K.

Reynolds Number	Aspect Ratio	Temperature Rise (K)
1000	0.125	1.69
1000	0.20	1.00
1000	0.25	0.76
1000	0.50	0.24
1000	0.75	0.12
1000	1.0	0.09

**Table 13 micromachines-14-01418-t013:** Summary of the maximum relative difference of dimensionless entrance length comparisons of velocity and *fRe* criteria at six aspect ratios.

Aspect Ratio	Reynolds	Max. Relative Difference
1.00	7	7%
0.75	7	10%
0.50	20	22%
0.25	100	53%
0.20	100	56%
0.125	3	64%

**Table 14 micromachines-14-01418-t014:** $L_{h}^{+}$ for fully developed laminar flow in rectangular channels.

α	Wiginton and Dalton [24]	Fleming and Sparrow [25]	Han [26]	McComas [27]	Beavers et al. [28]	Goldstein and Kried [29]
1	0.09	-	0.0752	0.0328/0.0324 [d]	0.030 [e]	0.090
0.75	-	-	0.0735	0.0310	-	-
0.50	0.085	0.047 [a]/0.070 [b]/0.095 [c]	0.0660	0.0255	0.015 [e]	-
0.40	-	-	-	0.0217	-	-
0.25	0.075	-	0.0427	0.0147	-	-
1/6	-	-	-	0.0110	-	-
0.20	0.08	0.03 [a]/0.052 [b]/0.08 [c]	-	-	0.015 [e]	-
0.125	-	-	0.0227	0.00938	-	-
0.10	-	-	-	0.00855	0.015 [e]	-
0.05	-	-	-	0.00709	0.015 [e]	-
0	-	0.0083 [a]/0.0090 [b]	0.0099	0.00588	-	-

[a] Used the fully developed flow condition: *K*(*z*)/*K*(∞) = 95%. [b] Used the fully developed flow condition: *U*/*U_fd_* = 95%. [c] Shah and London [22] interpreted their results [25] to use the fully developed flow condition: *U*/*U_fd_* = 99%. [d] Shah and London [22] presented the value. [e] Used the fully developed flow condition: *K*(*z*)/*K*(∞) = 95%.

**Table 15 micromachines-14-01418-t015:** Entrance length correlation Equation (24) coefficients based on velocity criterion: *U*/*U_fd_* = 99%.

Aspect Ratio	*A*	*B*	*C*	Mean Error	Max. Error	R^2^
1.000	0.707	0.08380	0.0733	1.25%	3.02%	1.000
0.750	0.745	0.07430	0.0745	1.50%	3.21%	1.000
0.500	0.905	0.05710	0.0810	2.42%	5.17%	1.000
0.250	1.340	0.01900	0.0795	4.46%	9.64%	1.000
0.200	1.520	0.00737	0.0683	4.67%	10.28%	0.999
0.125	1.960	0	0.0445	3.02%	6.88%	0.999

**Table 16 micromachines-14-01418-t016:** Hydrodynamic entrance length correlation coefficients Equation (24) based on *fRe* criterion: *fRe*/*f_z_Re_z_* = 99%.

Aspect Ratio	*A*	*B*	*C*	Mean Error	Max. Error	R^2^
1.00	0.665	0.09710	0.0698	1.50%	3.54%	1.000
0.75	0.679	0.08560	0.0684	1.51%	3.49%	1.000
0.50	0.729	0.05750	0.0614	2.15%	4.39%	1.000
0.25	0.788	0.02480	0.0380	2.09%	4.31%	1.000
0.20	0.777	0.02130	0.0321	1.82%	3.76%	1.000
0.125	0.685	0.01670	0.0237	1.75%	3.26%	1.000

## Data Availability

The data presented in this study are available on request from the corresponding author.

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
