# Peer review of "Simulations of Flows via CFD in Microchannels for Characterizing Entrance Region and Developing New Correlations for Hydrodynamic Entrance Length"

_micromachines, 2023, doi:10.3390/mi14071418_

Round 1

Reviewer 1 Report

Title:  Simulations of Flows via CFD in Microchannels for Charac-terizing Entrance Region and Developing New Correlations for Hydrodynamic Entrance Length

In this paper, the authors studied a device with microchannels that are relatively new and many correlations are not yet developed to design them efficiently. In microchannels, the flow regime is primarily laminar, where entrance length may occupy a significant section of the flow channel. Therefore, several computational fluid dynamic simulations were performed in this research to characterize the developing flow regime. The new correlations of entrance length were developed from a vast number of numerical results obtained from these simulations.

I recommended a major revision. The observations are:

1.      Modify the abstract section and write to the point. The last part is in the conclusion section part.

2.      The introduction section is weak and needs improvement. For the study of CFD and fluid, the author must consider the following results “Linear and quadratic convection on 3D flow with transpiration and hybrid nanoparticles, Nonlinear movements of axisymmetric ternary hybrid nanofluids in a thermally radiated expanding or contracting permeable Darcy Walls with different shapes and densities: Simple linear regression”.

3.      The reference must cite for the equations because these are not the authors' own equations.

4.      On page 3 line 55 what does means by steady-state?

5.      Fluid is incompressible please verify Eq. (2a).

6.       Results and discussion sections section must be physically improved.

7.      The English and grammar correction of the paper.

Minor editing of English language required

Author Response

  • The abstract section was modified slightly. The last two sentences of the old Abstract were removed as they appear under the conclusion section.
  • The Introduction section was rewritten for clarity with additional references.
  • Generally, a citation is not required if an equation is commonly known, but the authors have added additional citations.
  • Clarification was made on page 3. Now it reads

For the present study, the governing equations [25, 39] are conservation of mass Eq. (2) and conservation of momentum Eq. (3), assuming that gravitational force is negligible, fluid is incompressible, and the flow is under steady-state condition.

  • The authors of this paper are US citizens who are familiar with the English language and take care to avoid grammar and spelling mistakes. However, if the journal editors wish to go through their recommended language editors for improvement, the authors will be agreeable to paying a reasonable service fee.

Reviewer 2 Report

I appreciate the good work done by authors. However, I suggest improving and adding few things which I explain point wise.

1. Abstract must contain some major findings of the work.

2. The introduction. The authors should make it clear why this study is novel.

3. In the Introduction, the literature review was not logically organized and all literatures cited seem separate descriptions without connections. The readers can't know what the state-of-art methodologies or gaps the current study plans to resolve or fill, and how significant or what contribution the current study is?. Rewrite the last paragraph of introduction section and make it concise which contains the objective, novelty, motivation, and why you choose this problem. 

4. For validation of the governing equation 2(a) to 3(b), must cite a few references.

5.Improve the quality of the paper by fixing some typos errors.

6. Provide more information about the used numerical method.

7Provide the physical applications of the considered problem.

8. Must add future direction of your research..

Consequently, it is important to call your attention to the fact that all the sections must be logically connected. Based on the fact that the outcome of a research, after published, is cited and built-upon in the world, it is very important to improve the quality of presentation. Consequently, you need to
        (a) connect the title to the abstract
        (b) connect the abstract to the introduction
        (c) prepare the introduction to accurately lead to the analysis of result.
        (d) conclude the report based on the facts you have analyzed and discussed.

It is  fine.

Author Response

  • The significant findings of this work are two new correlations for hydrodynamic entrance length in microchannels, which are mentioned in the last line of the Abstract.
  • The Introduction section was rewritten for clarity with additional references.
  • Additional references were added for the governing equations
  • We have proofread the manuscript and fixed various grammatical errors.
  • Physical applications of the considered problem are described in the first paragraph of the Introduction section.
  • Future research must focus on developing new correlations for friction factor in the entrance region of microchannel.
  • The Title stresses new correlations for entrance length and is linked in the Abstract.
  • The abstract emphasizes CFD simulations and developing entrance length correlation.
  • The introduction describes macrochannels and microchannels and gives the analysis process.
  • The analysis and results in Section 4 are discussed clearly and in detail.

Reviewer 3 Report

Authors have studied to to characterize the developing flow regime in Entrance Region. A three-dimensional laminar flow for 37 Reynolds numbers (0.1, 0.2...1, 2...10, 20...100, 200…1000), primarily in low regime with water flow through six rectangular microchannels (aspect ratio: 1, 0.75, 0.5, 0.25, 0.2, 0.125) have been modeled conducting 222 simulations to characterize flow developments and ascertain progressive velocity profile shapes. In my opinion, paper has significant results to warrant for publication. But I cannot recommend this paper in this present form, therefore, my comments are given below.

1.       Introduction is not sufficient; it should be extended to show the latest research in this area.

2.       Objective of the paper should be included. Also,  research gaps should be identified to correlate the objectives.

3.       Table 1. Shows the literature, which is quite good. I urge to include numerical study also.

4.       Table 2 must be redraw for better understanding. The present form of the table is not understandable.

5.       Governing equations should be extanded in 3d forms.

6.       Boundary conditions should be presented in picture for better understanding.

7.       Check the equation 14(a).

8.       In figure 4a and 4b, present numerical results, natrajan and laxman results have been shown. But where is the McComes results.

Need extensive editing

Author Response

  • The Introduction section was rewritten for clarity with additional references.
  • The objective of this paper is described in Section 1.1. It justifies the new analysis required for microchannels to develop new correlations, which is the objective of this research.
  • Above Table 2, the following text was added for clarity.

From Table 2, several researchers explored hydrodynamic entrance length for rectangular channels, but none agreed due to choosing various aspect ratios and fully developed criteria, as shown in Table 14.

  • (3) in 3D form has been added.
  • A boundary condition figure could be added, but due to the MDPI formatting, that would require manual adjusting all the figure numbers and their references. If the editor or reviewer deems this necessary, the authors will comply.
  • Equation 14a is correct as written.
  • McComas’ results are incredibly close to those of Natarajan and Lakshmana, where they fall on each other. The following was added above Figure 4.

McComas and Natarajan & Lakshmana are stacked on top of each other.

Round 2

Reviewer 1 Report

Accept in present form

Minor editing of the English language required

Author Response

Reviewed the manuscript and made additional grammatical corrections.

Reviewer 3 Report

Authors have addressed most of the comments comments. But boundary conditions in pictures has not been presented. Boundary conditions in pictorial view will help to readers to understand. Therefore, it must be included.

Minor editing is required

Author Response

Added boundary condition figure in section 2.4.